# Prediction of Fatty Liver Disease in a Chinese Population Using Machine-Learning Algorithms

**DOI:** 10.3390/diagnostics13061168

**Published:** 2023-03-18

**Authors:** Shuwei Weng, Die Hu, Jin Chen, Yanyi Yang, Daoquan Peng

**Affiliations:** 1Department of Cardiovascular Medicine, The Second Xiangya Hospital, Central South University, Changsha 410011, China; wengshuwei@csu.edu.cn (S.W.); miraclehudie@163.com (D.H.); chenjin412@csu.edu.cn (J.C.); 2Research Institute of Blood Lipid and Atherosclerosis, Central South University, Changsha 410011, China; 3Health Management Center, The Second Xiangya Hospital, Central South University, Changsha 410011, China

**Keywords:** fatty liver disease, machine learning, XGBoost, early screening

## Abstract

Background: Fatty liver disease (FLD) is an important risk factor for liver cancer and cardiovascular disease and can lead to significant social and economic burden. However, there is currently no nationwide epidemiological survey for FLD in China, making early FLD screening crucial for the Chinese population. Unfortunately, liver biopsy and abdominal ultrasound, the preferred methods for FLD diagnosis, are not practical for primary medical institutions. Therefore, the aim of this study was to develop machine learning (ML) models for screening individuals at high risk of FLD, and to provide a new perspective on early FLD diagnosis. Methods: This study included a total of 30,574 individuals between the ages of 18 and 70 who completed abdominal ultrasound and the related clinical examinations. Among them, 3474 individuals were diagnosed with FLD by abdominal ultrasound. We used 11 indicators to build eight classification models to predict FLD. The model prediction ability was evaluated by the area under the curve, sensitivity, specificity, positive predictive value, negative predictive value, and kappa value. Feature importance analysis was assessed by Shapley value or root mean square error loss after permutations. Results: Among the eight ML models, the prediction accuracy of the extreme gradient boosting (XGBoost) model was highest at 89.77%. By feature importance analysis, we found that the body mass index, triglyceride, and alanine aminotransferase play important roles in FLD prediction. Conclusion: XGBoost improves the efficiency and cost of large-scale FLD screening.

## 1. Introduction

Fatty liver disease (FLD), a global epidemic disease, is an important risk factor for liver cancer [1,2]. However, the harm is not limited to the liver itself. Some studies have shown that fatty liver can significantly increase the incidence of fatal and non-fatal cardiovascular events [3], and FLD patients are more likely to be associated with obesity [4], hyperlipidemia [5], hypertension, and type 2 diabetes [6] than healthy people. Furthermore, as fatty liver disease progresses, the risk of CKD significantly increases with the degree of liver fibrosis [7]. The fatty liver disease (FLD) guideline for the Asian population [8] highlights that there is no national epidemiological survey on FLD in China, and the reported studies are mostly from economically developed regions, which may lead to some degree of bias in the epidemiological characteristics. Therefore, early FLD screening is not only necessary to reduce the socioeconomic burden of FLD, but also to improve the epidemiological investigation of FLD in China. For FLD, liver biopsy is undoubtedly the “gold standard” for diagnosis [9]. However, as a screening method, its high cost and invasive nature do not make it the first choice for fatty liver screening. Analogously, ultrasound, as an effective diagnostic method, relies on the operation and judgment of the ultrasound doctor. Therefore, an urgent need exists to develop cost-saving and non-invasive methods to screen fatty liver.

As a prediction tool, machine learning (ML) represents the latest development of statistics. Unlike the traditional statistical model, which depends on certain assumptions for data and a clear mathematical form, ML does not have any assumptions about the data, and the results eliminate the classical statistical framework based on hypothesis testing. The prediction efficiency of ML models based on algorithms or programs is high, and the results of cross-validation are easy to understand, so ML can be widely used in medical diagnosis trials today. Among them, several ML methods, such as random forest (RF), artificial neural network (ANN), K-nearest neighbor (KNN), and support vector machine (SVM) have played an important role in the prediction of many diseases.

Previous studies [10] have extracted the gray-value distribution features of children’s liver ultrasound images in a given region of interest, and then constructed an ML discriminant model of liver lesions by a variety of maximum-likelihood classification methods. The prediction accuracy is better than the traditional liver and kidney index and liver echo intensity attenuation index. Acharya et al. [11] extracted abdominal ultrasound features with the curvelet transform method, reduced features through locality-sensitive discriminant analysis, and used a probabilistic neural network classifier based on only six features to distinguish the normal liver, fatty liver, and liver cirrhosis with an accuracy of 97.33%, specificity of 100%, and sensitivity of 96%.

Based on in-depth ML approaches to diagnosing FLD, there are many artificial methods that can diagnose fatty liver with high accuracy. However, most of these diagnoses are based on abdominal ultrasound or computed tomography (CT) images, which are costly for fatty liver screening. Compared with the image-based ML screening methods, using physical examination data and blood biochemical indexes as predictive indicators can screen fatty liver in an efficient and economical way. Thus, the main purpose of this study was to build an efficient and robust FLD screening ML model based on these indicators.

## 2. Materials and Methods

### 2.1. Study Data

The dataset used in this study was provided by the health management center of the Second Xiangya Hospital of Central South University, Changsha, China, and included the data of 36,527 patients. We enrolled individuals aged 18–70 years from January 2013 to December 2019. During the process of data collection, no privacy information was included. Only 23 indexes, including physical examination data, age, and blood biochemistry indexes were included. Because ML models rely on data integrity, 4908 individuals with missing values of more than 30% were excluded, and the remaining missing values were completed by the multiple interpolation method. In this study, the diagnosis of FLD was based on the results of abdominal ultrasound images. The ultrasound machines used in this study were the Philips Medical Systems model iU22 and model Epiq (Philips Ultrasound, Bothell, WA, USA). All the diagnostic results of abdominal ultrasounds were performed by attending physicians in our hospital’s imaging center and were reviewed by senior physicians. The diagnostic criteria for FLD were based on the guidelines published by the Chinese Medical Association in 2010 [12]. The diagnosis of FLD was confirmed if at least two of the three following findings were present: diffuse echogenicity enhancement of the liver parenchyma in the near field, stronger than that of the kidney; poor visualization of intrahepatic duct structures; and the gradual attenuation of liver echogenicity in the far field. Therefore, we excluded 1045 individuals who had not yet completed the screening process. Finally, of the 30,574 individuals remaining in the study, 3474 of them were diagnosed with FLD by abdominal ultrasound (Figure 1). 

### 2.2. Data Processing

In machine learning data preprocessing, class imbalance is a common issue where the number of samples in one category is much larger than in the other. This can present challenges for machine learning models, and the severity of the imbalance depends on the proportion of samples in each category. To address this problem, this study used a synthetic minority over-sampling technique nominal continuous (SMOTE-NC) [13] to handle unbalanced class data. SMOTE-NC is an extension of the synthetic minority over-sampling technique (SMOTE) that can handle nominal and continuous features. In an imbalanced dataset, SMOTE-NC generates synthetic samples for the minority class by oversampling the existing samples using interpolation. When generating synthetic samples, SMOTE-NC takes into account both continuous and nominal features and ensures that the synthetic samples are representative of the underlying data distribution.

Feature selection is a crucial aspect of classification tasks, as it can significantly impact the performance of the model. The main objective of feature selection is to identify the most relevant subset of features that can improve the accuracy of classification. In the context of the 11 characteristic variables presented in Table 1, the selection of these variables was based on several factors. These included identifying the most commonly used variables for predicting FLD, adding additional variables to increase the variety of features, and using a stepwise backward selection method based on Akaike Information Criterion (AIC) [14]. AIC is a powerful tool for assessing the performance of a model in terms of both its predictive accuracy and its complexity. It is founded on the concept of entropy, which enables it to capture the trade-off between these two competing factors. By comparing the AIC values of different models, researchers can identify the most effective and parsimonious one for their purposes. In order to filter features and avoid multicollinearity, this study utilized the reverse stepwise-regression algorithm based on AIC. This involved introducing all variables into an equation, and then iteratively deleting the variable that maximized the AIC value until the minimum AIC value was reached. The resulting variables and their corresponding AIC values are shown in Table 2. To automate the feature selection process, a R program was used to compare the AIC values of candidate variables and include those that contributed to the model. The variables listed in Table 2 reflect the remaining variables that were found to contribute to the machine learning model, and the corresponding AIC value indicates the change in AIC value after adding each variable. 

We used the createDataPartition function in the caret package to divide the training set and the test set, in which the test set accounted for 70% of all data. Continuous variables were normalized by subtracting the average value and dividing by the standard deviation. To overcome the imbalance problem in the training set, we used the synthetic minority oversampling technique to randomly generate new individuals of the minority, which have similar features to the original individuals of the minority class. 

### 2.3. Establishment of the Model 

We used the eight most common classifiers to build an FLD screening model. As one of the most commonly used generalized linear regression models for binary data, logistic regression (LR) can not only provide prediction results, but also indicate the weight of each independent variable in the prediction. RF is a classifier that integrates multiple decision trees. All decision trees are independent of each other, and each decision tree splits the maximum information gain, and finally outputs the results of classification after reaching the threshold; the RF results are based on the majority of all decision trees. As a common two-classification model, the SVM maps the feature vector to the space, and finds the separation hyperplane with the largest interval in the feature space. This approach makes the classification results more robust and improves the generalization ability of the model. Linear discriminant analysis (LDA) is a classical supervised learning method based on data dimensionality reduction, which classifies the data by projecting the data from a high-dimensional space to a lower-dimensional space and ensuring that the intra-class variance of each class is small and that the mean difference between classes is large. Quantitative descriptive analysis (QDA) is a variant of LDA that allows the nonlinear separation of data. KNN is regarded as a nonparametric, lazy algorithm model that is based on adjacent samples with the minimum Euclidean distance. This model makes no assumptions about the data, and there is no clear training data process. Because it assigns the same weight to different features, the model is easily affected by noise. Extreme gradient boosting (XGBoost) is an improved boosting algorithm based on the gradient boosted decision tree (GBDT) method. Unlike with classical GBDT, second-order Taylor expansion is used in XGBoost on the error part of the loss function, which improves the accuracy of the loss function definition. Because of the L2 regularization in the cost function, the complexity of the XGBoost model is controlled, which greatly reduces the possibility of overfitting. Because of these characteristics, XGBoost has excellent classification and regression prediction performance. An artificial neural network (ANN) is a black box model constructed by simulating the brain’s neural structure, and is generally composed of an input layer, hidden layer, and output layer. Each layer may contain multiple neurons. The number of neurons in the input layer depends on the input parameters, and the number of neurons in the other layers is adjusted according to the actual situation. In this model, the input parameters are connected to the neuron on the basis of a certain weight. The activation threshold of the neuron is determined by setting the activation function; then, the signal is further transmitted in the network. The neural network can achieve self-learning through forward propagation or back propagation, and gradually optimizing the weights and deviation values in the process until the value of the loss function tends to be stable and reaches the expected value. Finally, the network generates the results through the output layer. 

In this study, all the parameters in the models were adjusted by cyclic traversal, and the highest area under the curve (AUC) value was regarded as the selection standard of the model parameters (Figure 2). A 10-fold cross validation was carried out to estimate the performance of each model.

### 2.4. Model Performance Assessment

We verified the predictive ability of the ML model by calculating the area under the curve (AUC), accuracy, sensitivity, specificity, positive predictive value, negative predictive value, and kappa value. In this section, we provide an overview of the various metrics used to evaluate the performance of machine learning models.

Accuracy is a crucial evaluation metric in machine learning and represents the proportion of correctly predicted samples in the overall sample. A higher accuracy indicates better classification performance. The formula used for calculating accuracy is as follows:*Accuracy = (TP + TN)/(TP + FP + TN + FN) × 100%*
where *TP*, *TN*, *FP*, and *FN* represent true positive, true negative, false positive, and false negative, respectively.

Sensitivity, as known by the true positive rate, is a critical performance metric for machine learning models, as it quantifies the model’s ability to accurately identify patients who test positive. It measures the proportion of actual positive cases that the model correctly identifies as positive, providing insight into the model’s ability to detect true positives.
*Sensitivity = TP/(TP + FN) × 100%*

Specificity refers to the proportion of negative cases identified out of all negative cases. It is a measure of the ability of a machine learning model to correctly identify negative cases. The higher the specificity, the lower the false positive rate, and the more accurate the model’s negative predictions.
*Specificity = TN/(TN + FP) × 100%*

Positive predictive value (PPV) is a performance metric in machine learning that measures the proportion of true positive predictions made by the model among all positive predictions. In other words, PPV represents the probability that a positive prediction is actually correct.
*PPV = TP/(TP + FP) × 100%*

Negative predictive value (NPV) is another performance metric in machine learning that measures the proportion of true negative predictions made by the model among all negative predictions. NPV represents the probability that a negative prediction is actually correct.
*NPV = TN/(TN + FN) × 100%*

Kappa is a statistical measure of agreement that takes values between −1 and 1. In the context of the classification problem under study, it indicates the degree of agreement between the model’s predicted results and the actual classification results. As a rule of thumb, a higher kappa value is often regarded as indicative of stronger agreement between the classifier and the actual results.
*Kappa = (P_o_ − P_e_)/(1 − P_e_)*
where *P_o_* is the observed proportion of agreement between the two classifiers, and *P_e_* is the expected proportion of agreement due to chance.

Receiver operating characteristic (ROC) curves are a graphical representation of the performance of a binary classifier system. The ROC curve is created by plotting the true positive rate (TPR) against the false positive rate (FPR) at various threshold settings. A perfect classifier has an ROC curve that passes through the top left corner of the plot, indicating a high TPR and low FPR. AUC is a measure of the classifier’s ability to distinguish between positive and negative classes, with an AUC of 1.0 indicating perfect classification and an AUC of 0.5 indicating random guessing.

The Shapley value is a concept from cooperative game theory that measures the marginal contribution of each player to a cooperative game. For machine learning, the Shapley value represents the contribution of a feature to a prediction by considering all possible combinations of features that could have been used in the model. It measures the average change in the model’s output when a feature is added, compared to when the feature is not included. The Shapley value can help identify the most important features for a particular prediction and to understand how the model makes decisions. While the SHAP library’s kernel Explainer is capable of computing Shapley values for any machine learning model, it is computationally inefficient for KNN models. Although the k-means clustering algorithm can be used to summarize the data and improve computational efficiency, this comes at the expense of the model’s accuracy. To address this, we evaluated the feature importance of the KNN model using root mean square error loss after permutation in this study.

## 3. Results

### 3.1. Features of Participants

A total of 30,574 participants were finally included in this study, including 14,250 males and 16,324 females. Among the participants, 3474 participants were diagnosed with fatty liver; their average age was 45.7 ± 10.9 years, whereas the age of the individuals without FLD was 41.7 ± 12.0 years. All variables we included are shown as mean (SD), and the two-sample *t*-test showed a significant difference between the two groups (Table 3). The density distribution curves of all included characteristic variables are shown in Figure 3.

### 3.2. Model Performance

The ROC curves for all models are shown in Figure 3. After comparing the selected ML models, we found the following: (1) The prediction accuracy of XGBoost for FLD was 89.7%, and it had high AUC, sensitivity, and specificity. (2) The prediction ability of SVM was closest to that of XGBoost, and its sensitivity was better than that of XGBoost, suggesting that SVM is also one of the best prediction models. (3) The kappa values of XGBoost and SVM were both higher than 70%, which indicates their good repeatability. (4) Although the accuracy of RF and KNN was close to 75%, their positive predictive values and kappa values were low, suggesting that the models based on these two algorithms have low positive prediction efficiency and poor repeatability (Table 4). According to the results of feature importance analysis, we found that BMI, ALT, and Tg play important roles in all models (Figure 4). 

## 4. Discussion

This study included the clinical data of 30,572 subjects and 8 ML models, which makes it by far the largest machine learning study based on physical examination and blood biochemical indicators to predict fatty liver in the Chinese population. According to the results, it is not difficult to find that ML can efficiently predict the occurrence of FLD, and the XGBoost model is the best predictor among all the analyzed models. This is likely due to the XGBoost model’s ability to adaptively adjust the depth of trees and weights of leaf nodes to minimize the loss function, as well as mitigate overfitting issues by incorporating regularization terms. The XGBoost model can handle datasets with a large number of features and samples, and identify key factors through feature importance evaluation, thereby improving the model’s interpretability and reliability. These advantages make the XGBoost model highly accurate in binary classification predictions and perform exceptionally well in the diagnosis of many clinical conditions [15,16,17]. Patients with fatty liver disease frequently exhibit metabolic syndrome characteristics, such as overweight, insulin resistance, and atherogenic dyslipidemia that are characterized by elevated plasma triglyceride concentrations. Research indicates that the prevalence of NAFLD among obese patients is as high as 80%, compared to 16% in individuals with a normal body mass index and no metabolic risk factors [18,19]. Dyslipidemia in patients with FLD is primarily characterized by hypertriglyceridemia due to large very-low-density lipoprotein particles, increased levels of small and dense low-density lipoprotein particles, and reduced high-density cholesterol levels [20,21]. This alteration in lipid profile can be attributed to heightened cholesteryl ester transfer protein activity [22]. During the natural progression of fatty liver disease, liver enzyme levels also fluctuate, and around 20% of patients with NAFLD have substantial changes in liver enzyme levels, with aspartate aminotransferase (AST) and alanine aminotransferase (ALT) levels remaining within the normal range or being modestly elevated (1.5–2 times the upper limit of normal) [23]. ALT is considered an essential indicator of liver inflammation and a significant marker of disease amelioration. A recent study corroborated that serum ALT levels are an effective indicator of histological changes and can be utilized as an efficient treatment indicator [24]. These observations support the significance of BMI, ALT, and triglycerides in most models in this study.

Moreover, on the basis of the decision tree model, we constructed a simplified screening model (Figure 5) for physicians to evaluate FLD in the absence of imaging and pathological evidence, which is helpful for the preliminary screening of patients with fatty liver.

The traditional diagnosis of fatty liver mainly depends on imaging results or invasive biopsies, which have high medical and human resource requirements. However, the predictive index of FLD screening based on machine learning is easier to obtain, and the results do not rely on the subjective judgments of doctors. The dataset used for predicting fatty liver disease is mainly unbalanced and categorical, with a much lower number of patients with fatty liver disease than those without. This study utilized the SMOTE-NC method to preprocess the unbalanced data and applied the AIC backward propagation technique for feature engineering. As a result, our model achieved an accuracy comparable to that of a previous study while using fewer feature variables [25,26], including demographic indicators, blood glucose, liver function test, and blood lipid profiles. This narrower range of features reduced the dimensionality issue caused by having too many features, making it easier to collect data from primary medical institutions. Lipid deposition and fibrosis commonly coexist in the liver during the course of FLD. Among them, fibrosis becomes more representative when liver dysfunction reaches the end stage [27]. Samir Hassoun et al. [28] developed a machine learning model based on the general population in the United States that can effectively screen for severe liver fibrosis features. This complements our research well and provides a more comprehensive model coverage for screening for FLD. Therefore, ML model-based screening can be carried out in most basic medical institutions and provides new insight into doctors’ diagnoses. Because of the high robustness and prediction accuracy of the model, there is no doubt that ML is feasible for large-scale FLD screening.

This study had some limitations. First, although more than 30,000 samples were included in this study, these samples were all from the Second Xiangya Hospital of Central South University, and the population representation was less comprehensive than that of multi-center clinical studies. The generalization ability of the model in different ethnic groups is open to question. Second, the response variables used in the machine learning model constructed in this study are solely based on the diagnostic results of abdominal ultrasound, which have a lower level of evidence compared to liver biopsy and magnetic resonance imaging (MRI). This may potentially affect the accuracy of the predictions. Thirdly, tumor, hepatitis, and other metabolic diseases (such as diabetes, hyperthyroidism, etc.) were not excluded from the population included in this study. As a result, the potential impact of these factors on the predictive model could not be fully assessed. Lastly, it should be noted that we did not gather data on alcohol consumption and medication history among the study population. Therefore, we were unable to rule out the potential interference caused by alcohol and drugs. Previous studies have indicated that both factors can influence the development of fatty liver [29,30,31]. To improve our model, future studies should gather more detailed information on alcohol intake and medication history. Even so, the ML model based on the XGBoost algorithm still had an accuracy of nearly 90% and its AUC value reached 96%; thus, it can play an important role in large-scale FLD screening.

## 5. Conclusions

In this study, we found that ML models, especially XGBoost, can predict FLD through demographic indicators, blood glucose, liver function test, and blood lipid profile with high accuracy and good repeatability. Among 11 predictive indicators, BMI, Alt, and Tg are vitally important for most models. With the aid of these machine learning methods, physicians can now evaluate a patient’s fatty liver condition in its incipient stages, even in the absence of liver biopsy or imaging evidence. This early assessment can facilitate timely diagnosis and provide a novel avenue for the early screening of fatty liver in primary healthcare facilities. In future research, we will collect more data from various populations to further modify the model and give it better generalization ability.

## Figures and Tables

**Figure 1 diagnostics-13-01168-f001:**
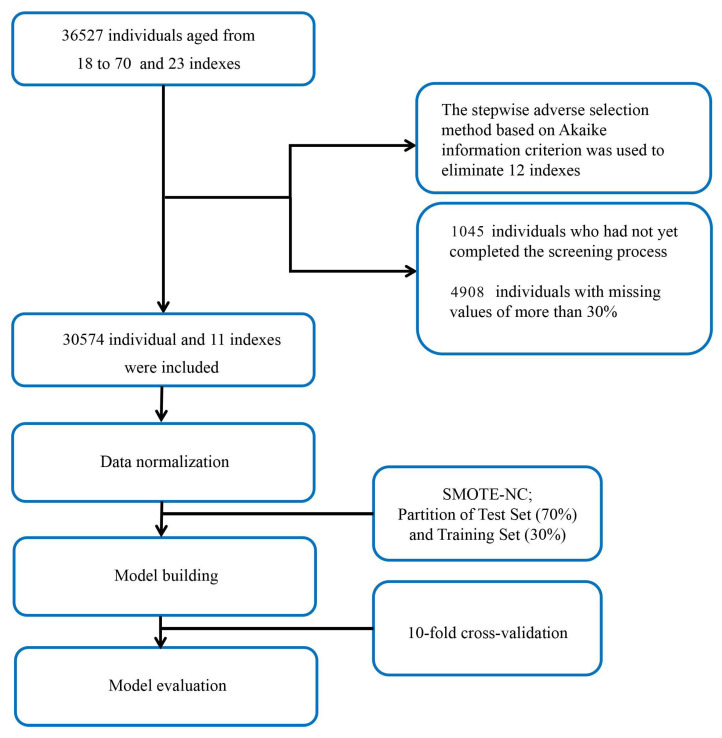
Data processing flowchart.

**Figure 2 diagnostics-13-01168-f002:**
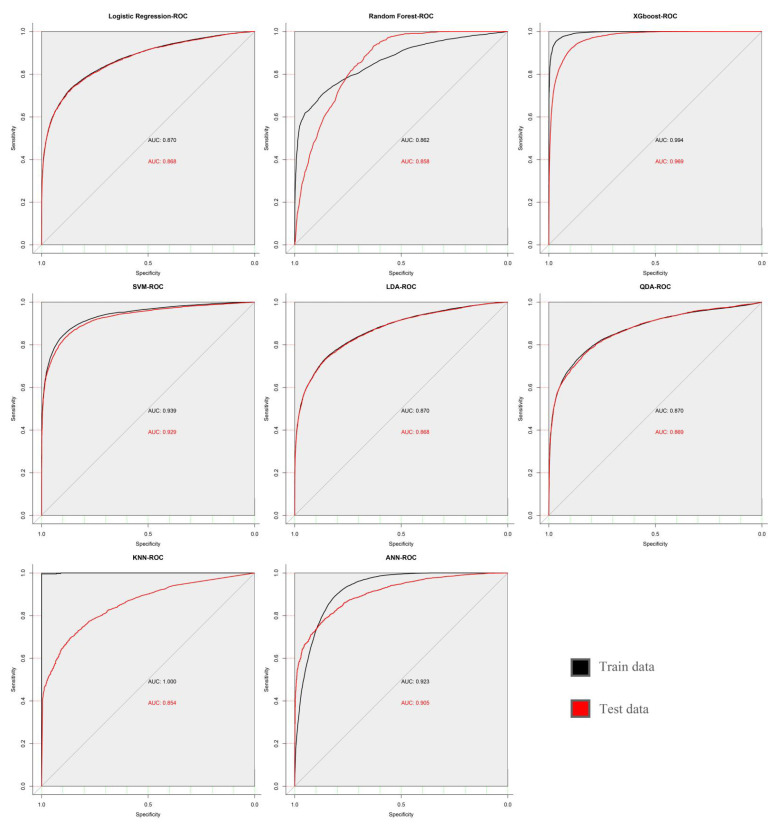
Receiver operating characteristic curve of all machine learning models.

**Figure 3 diagnostics-13-01168-f003:**
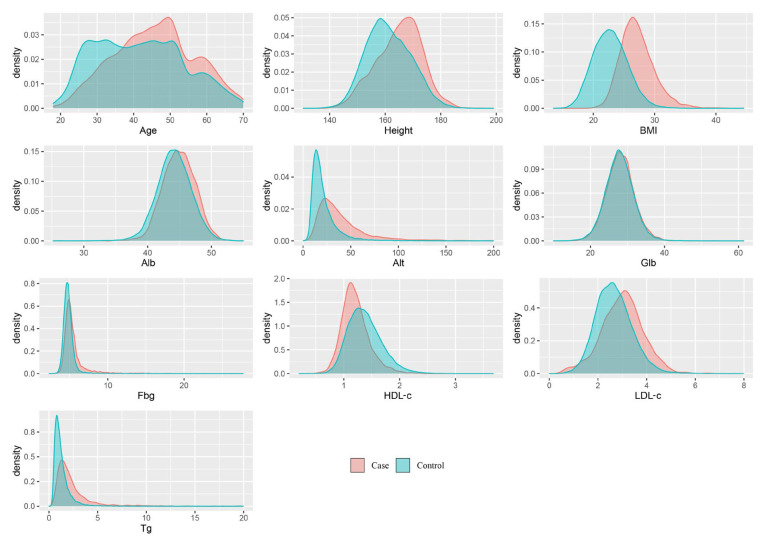
The density distribution of all predictive variables. BMI: body mass index; Alb: albumin; Alt: glutamic-pyruvic transaminase; Glb: globulin; Fbg: fasting blood-glucose; HDL-c: high-density lipoprotein cholesterol; LDL-c: low-density lipoprotein cholesterol; Tg: triglyceride.

**Figure 4 diagnostics-13-01168-f004:**
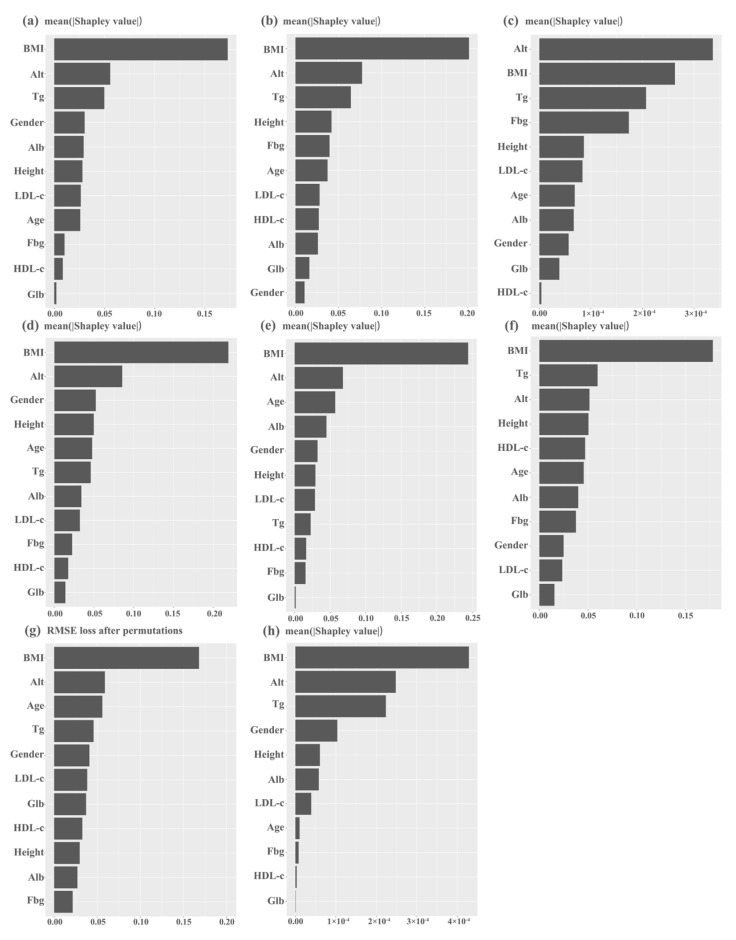
Feature importance analysis of all machine learning models: (**a**) logistic regression; (**b**) random forest; (**c**) extreme gradient boosting; (**d**) support vector machine; (**e**) linear discriminant analysis; (**f**) quantitative descriptive analysis; (**g**) k-nearest neighbor; and (**h**) artificial neural network.

**Figure 5 diagnostics-13-01168-f005:**
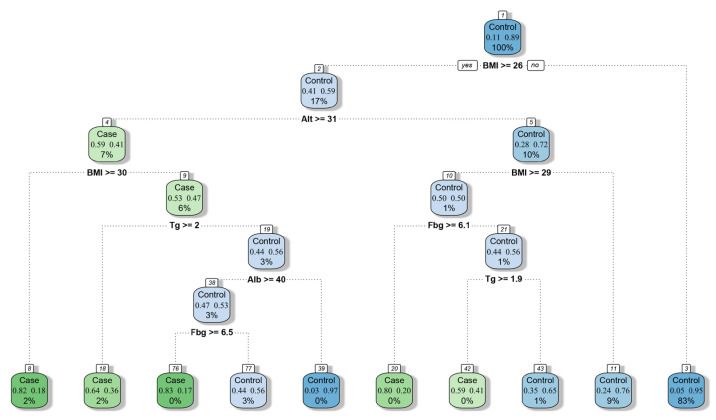
Decision tree for screening fatty liver disease.

**Table 1 diagnostics-13-01168-t001:** Detailed variable description.

Type of Data	Variable	Description
Demographics Data	Gender	Gender of the participant
	Age	Age in years at screening
Examination Data	Height	Height of the participant
	Weight	Weight of the participant
	BMI	Body Mass Index
	Sp	Systolic pressure
	Dp	Diastolic pressure
Laboratory Data	TC	Total cholesterol
	Tg	Triglyceride
	LDL-c	Low-density lipoprotein cholesterol
	HDL-c	High-density lipoprotein cholesterol
	Fbg	Fasting blood glucose
	Alb	Albumin
	Glb	Globulin
	Tp	Total protein
	DBil	Direct Bilirubin
	TBil	Total bilirubin
	Tba	Total bile acid
	ALT	Alanine aminotransferase
	AST	Aspartate aminotransferase
	BUN	Blood urea nitrogen
	Cr	Creatinine
	UA	Uric acid

**Table 2 diagnostics-13-01168-t002:** Feature Selection Based on Akaike Information Criterion.

Variable	Deviance	AIC
BMI	16,005	16,043
Alt	13,589	13,627
Tg	13,425	13,463
Fbg	13,356	13,394
Alb	13,263	13,301
Age	13,260	13,298
Ldl-c	13,237	13,275
Age	13,228	13,266
Glb	13,211	13,249
Alb	13,202	13,240
Hdl-c	13,187	13,225

BMI: body mass index; Alb: albumin; ALT: alanine aminotransferase; Glb: globulin; Fbg: fasting blood-glucose; HDL-c: high-density lipoprotein cholesterol; LDL-c: low-density lipoprotein cholesterol; Tg: triglyceride.

**Table 3 diagnostics-13-01168-t003:** Characteristics of study population.

Characteristic	Case (N = 3474 ^1^)	Control (N = 27,100 ^1^)	*p*-Value ^2^
Height	165.0 (8.1)	161.3 (8.0)	<0.001
Age	45.7 (10.9)	41.7 (12.0)	<0.001
Gender			<0.001
Female	823 (23.7%)	15,501 (57.2%)	
Male	2651 (76.3%)	11,599 (42.8%)	
BMI	27.2 (2.8)	22.8 (2.8)	<0.001
Alb	44.9 (2.6)	44.2 (2.7)	<0.001
ALT	40.7 (30.3)	22.2 (26.1)	<0.001
Glb	28.1 (3.7)	27.9 (3.8)	<0.001
Fbg	5.5 (1.8)	4.9 (1.0)	<0.001
HDL-c	1.2 (0.3)	1.4 (0.3)	<0.001
LDL-c	3.0 (0.9)	2.7 (0.7)	<0.001
Tg	2.5 (2.7)	1.3 (1.1)	<0.001
^1^ Mean (SD); n (%)			
^2^ Welch Two Sample *t*-test; Fisher’s exact test			

Data are represented as mean (SD) or number (proportion). BMI: body mass index; Alb: albumin; ALT: alanine aminotransferase; Glb: globulin; Fbg: fasting blood-glucose; HDL-c: high-density lipoprotein cholesterol; LDL-c: low-density lipoprotein cholesterol; Tg: triglyceride.

**Table 4 diagnostics-13-01168-t004:** Assessment of eight machine learning models.

	Accuracy	Sen	Spe	Ppv	Npv	AUC	Kappa
**XGBoost**	0.8977	0.9247	0.889	0.7272	0.9736	0.969	0.745
**SVM**	0.8586	0.8589	0.8959	0.8251	0.8213	0.929	0.7177
**ANN**	0.8116	0.9019	0.7827	0.5704	0.9614	0.913	0.5716
**LR**	0.7926	0.8565	0.7354	0.7439	0.851	0.868	0.5873
**LDA**	0.7903	0.8513	0.7356	0.7429	0.8465	0.868	0.5825
**QDA**	0.7887	0.8602	0.7245	0.737	0.8524	0.869	0.5797
**KNN**	0.7536	0.7543	0.7535	0.2817	0.9599	0.854	0.2933
**RF**	0.7322	0.83493	0.71907	0.27584	0.97142	0.858	0.2941

Various machine learning models are arranged according to the accuracy of FLD prediction. AUC, area under curve of test set; Sen: sensitivity; Spe: specificity; Ppv: positive predictive value; Npv: negative predictive value; XGBoost: extreme gradient boosting; SVM: support vector machine; ANN: artificial neural network; LR: logistic regression; LDA: linear discriminant analysis; QDA: quantitative descriptive analysis; KNN: k-nearest neighbor; RF: random forest.

## Data Availability

The data that support the findings of this study are available on request from the corresponding author, upon reasonable request.

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
