# Peer review of "Prediction of Fatty Liver Disease in a Chinese Population Using Machine-Learning Algorithms"

_diagnostics, 2023, doi:10.3390/diagnostics13061168_

Round 1
Reviewer 1 Report
In the present manuscript the authors report the results of a cross-sectional study investigating the ability of a machine learning model to identify fatty liver disease. The topic is certainly of interest given the epidemiology of this condition.
I have the following comments:
1) The authors state that “early FLD screening is urgently needed” and that “liver biopsy and abdominal ultrasound, the preferred FLD diagnosis methods, are not suitable for large-scale FLD screening in primary medical institutions”. I find these sentences problematic for two reasons. Firstly, screening for FLD in the general population is not recommended by international guidelines at this stage. Secondly, liver ultrasound is an easily applicable technique to evaluate the presence of steatosis in patients at risk (e.g. obese individuals, patients with diabetes or metabolic syndrome).
2) Please briefly describe the study population in the abstract and mention the technique that was used to identify steatsosis.
3) When discussing the extra-hepatic manifestations of NAFLD, one should also mention CKD (see doi: 10.3390/biom12010105)
4) Please provide more information on whether other forms of liver disease were excluded and how. Please also report how many radiologists were involved and which criteria were used for defining steatosis.
5) Use of an imperfect diagnostic tool as reference (compared with liver biopsy or magnetic resonance spectroscopy) should be mentioned as a limitation of the present study.
6) Apart from steatosis, the prognosis of patients with fatty liver is more dependent on the degree of liver fibrosis. This aspect should be stressed with reference to a recent study using AI to identify advanced liver fibrosis instead of steatosis (doi: 10.1016/j.ijmedinf.2022.104932)
Author Response
Dear Reviewer,
Thank you for taking the time to review our manuscript and for providing us with
your valuable feedback. Your constructive comments have helped us to improve the quality of our work. We appreciate your thoughtful and thorough review of our
manuscript.
We have carefully considered each of your suggestions and have made the following revisions:
1. Although international guidelines do not recommend screening for FLD in the general population, the Asian-Pacific Association for the Study of the Liver (APASL) guidelines for FLD in Asian populations (DOI: 10.1007/s12072-020-10094-2) state that there is a lack of national epidemiological surveys in China, and most reported studies are from economically developed regions, which may not represent the epidemiological characteristics of FLD in the entire country. Therefore, FLD screening in China is still necessary. While ultrasound is a widely accepted technique for evaluating liver fat deposition worldwide, the standardized training of ultrasound physicians for FLD diagnosis remains a demanding requirement for some economically underdeveloped rural medical institutions. In contrast, population data, physical examination data, blood lipids, and liver function indicators are more easily obtainable. There are now convenient mobile devices that can quickly detect the blood biochemical indicators included in the model, and the results do not require subjective judgment from ultrasound physicians. We have modified the wording based on your suggestion to explain the necessity of large-scale screening for FLD and the significance of using machine learning as a screening tool (see lines 17-20 and 50-55 of the revised manuscript).
2. Thank you for your suggestion. We have now incorporated your feedback into the abstract by briefly describing the study population and the techniques used for identifying FLD (see lines 23-26 of the revised manuscript).
3. Based on your suggestion, we have enhanced the discussion on the extra-hepatic manifestations of fatty liver disease in the revised manuscript. In addition to the cardiovascular and endocrine systems, the impact of fatty liver on the renal system is also critical. We have referenced the systematic analysis you provided on the association between liver cirrhosis and chronic kidney disease. Please refer to lines 49-50 of the revised manuscript for more details.
4. In this study, we did not include patients' medical history, medication use, or alcohol consumption information. Therefore, we cannot rule out the possibility that fatty liver may have been caused by medication, alcohol, or other metabolic-related diseases. As a result, we discussed these limitations in our revised manuscript, specifically in lines 376-384. However, despite these limitations, our machine learning model constructed using XGBoost still achieved an 89.77% prediction accuracy, indicating a high level of confidence in our model. Following your suggestion, we have provided detailed information on the ultrasound equipment used, personnel involved in diagnosis, and diagnostic criteria referenced, along with relevant references (see lines 98-106 of the revised manuscript).
5. Thank you for your feedback. The supervised machine learning model constructed in this study was based on ultrasound reports, and its level of evidence is lower than that of liver biopsy and magnetic resonance imaging. Based on your suggestion, we have added this research limitation to the discussion section of the manuscript. Please refer to lines 372 to 376 of the revised manuscript for more details.
6. Your suggestion is very appropriate. The progression of liver fibrosis has a significant impact on the development of fatty liver disease, often co-occurring with lipid deposition in the liver. In this study, the diagnostic criteria for the fatty liver disease were based on the Chinese Medical Association's 2010 ultrasound diagnostic criteria, which include both lipid deposition and fibrosis in the course of the disease. As fatty liver disease progresses to its end stage, liver fibrosis becomes more representative than lipid deposition, making screening for advanced liver fibrosis more meaningful. Therefore, we have added the literature you suggested to our discussion and believe that the article can complement our research on fatty liver disease diagnosis. Please refer to lines 356 to 361 of the revised manuscript for more details.
Once again, we sincerely appreciate your time and effort in reviewing our manuscript. Please let us know if you have any further comments or suggestions.
Best regards,
Shuwei Weng

Reviewer 2 Report
The study is well presented using a standard ML pipeline. The samples are sufficient to produce significant results.
Some limitations/suggestions
1. I find Figure 4 : Feature Importance Analysis to be unclear ,, It doesn't specify the features clearly and the difference using different algorithms is not represented.
2. The conclusions stated the this method could be used to replace clinical methods such as biopsy and ultrasound. This statement should be reworded. AI and ML are not developed to replace but to assist clinicians in making decisions.
3. it is not discussed clearly why XGboost yielded higest accuracy. at least cite some references to validate your findings.
Author Response
Dear Reviewer,
Thank you for taking the time to review our manuscript and for providing us with
your valuable feedback. Your constructive comments have helped us to improve the quality of our work. We appreciate your thoughtful and thorough review of our
manuscript.
We have carefully considered each of your suggestions and have made the following revisions:
1. Based on your suggestion regarding Figure 4, we have re-drawn the images depicting the analysis of feature importance for different models. We have improved the labeling size of the different features in the images, making them more legible and easier to read.
2. Indeed, this method is intended solely for screening fatty liver and providing diagnostic suggestions for clinical workers in the early diagnosis of fatty liver. It is not intended to replace ultrasound or histopathological examination completely. I have revised this statement in the conclusion section. Please refer to lines 391-395 of the revised manuscript for more details.
3. Your suggestion was very constructive. After reviewing existing diagnostic studies related to machine learning, we found that the XGBoost model not only performs well in screening for the fatty liver but also shows excellent performance in predicting other diseases. This indirectly reflects the feasibility of the XGBoost model in binary clinical diagnosis. In the discussion, we followed your suggestion and explained the reasons for its high accuracy in this study based on the characteristics of the XGBoost model. We also included relevant reviews and similar cross-sectional studies to support our viewpoint. Further information can be found in lines 314-321 of the revised manuscript.
We hope that these changes have addressed your concerns and have improved the
clarity and quality of our manuscript.
Once again, we sincerely appreciate your time and effort in reviewing our
manuscript. Please let us know if you have any further comments or suggestions.
Best regards,
Shuwei Weng

Round 2
Reviewer 1 Report
I congratulate the Authors for their thorough review. I have no further comments.